# Controversies Regarding Mesh Utilisation and the Attitude towards the Appendix in Amyand’s Hernia—A Systematic Review

**DOI:** 10.3390/diagnostics13233534

**Published:** 2023-11-26

**Authors:** Dan Bratu, Alin Mihetiu, Alexandra Sandu, Adrian Boicean, Mihai Roman, Cristian Ichim, Horatiu Dura, Adrian Hasegan

**Affiliations:** 1County Clinical Emergency Hospital of Sibiu, 550245 Sibiu, Romania; dan.bratu@ulbsibiu.ro (D.B.); alexandrasandu96@yahoo.ro (A.S.); adrian.boicean@ulbsibiu.ro (A.B.); mihai.roman@ulbsibiu.ro (M.R.); cristian.ichim@ulbsibiu.ro (C.I.); horatiu.dura@ulbsibiu.ro (H.D.); adrian.hasegan@ulbsibiu.ro (A.H.); 2Faculty of Medicine, Lucian Blaga University of Sibiu, 550169 Sibiu, Romania

**Keywords:** vermiform appendix, inguinal hernia, surgical treatment, mesh, laparoscopy

## Abstract

Inguinal hernia containing the vermiform appendix is a rare entity. It is more common in children than in adults. It can be discovered incidentally during the surgical intervention performed for the cure of the inguinal hernia or when the appendix shows inflammatory changes, a situation that can lead to diagnostic confusion with a number of other diseases. Imaging can guide the diagnosis, which often comes as an intraoperative surprise. The therapeutic approach is controversial both in terms of whether or not to perform an appendectomy in the case of an appendix without inflammatory changes and especially in terms of using a mesh during the hernia repair process. Since the pathology is not very frequent, there are no standardized stages in terms of surgical ethics that can guarantee good surgical practice. The study aimed to carry out a review of the specialized literature to obtain some conclusions or trends regarding the management of this pathology. The low frequency of this type of hernia did not allow the consultation of large-scale studies or extensive reviews focusing on case reports or case series communications. The obtained results were statistically analyzed and integrated in relation to the surgical attitude depending on the particularities of the condition.

## 1. Introduction

Hernia represents the protrusion of the abdominal viscera through an anatomically predictable opening. It is considered that, except for the pancreas, all intra-abdominal organs can be located inside a hernia sac. Of these, one of the rarest found is the vermiform appendix.

This type of hernia got its name thanks to Claudius Amyand, a surgeon of French origin who practiced in England. In 1735, he performed the first appendectomy on an 11-year-old patient presenting with acute appendicitis in a right inguinal hernia sac. The patient survived, and the intervention also coincided with the first successful appendectomy. Amyand subsequently described the operation in a paper for the Royal Society. 

Over time, many other surgeons took credit for the first appendectomy, not recognizing or knowing Amyand’s description. In the early 20th century, John Blair Deaven gives him the credit he deserves by proving that the first appendectomy (Amyand’s) had taken place 150 years earlier than originally thought. The eponym Amyand’s Hernia was used in 1953 by Creese and later by Hiatt and Hutchinson, a fact that gave due credit to the first surgeon who operated on this particular type of condition [1,2].

Inguinal hernia repair is one of the most common surgical interventions in general surgery. Annually over 20 million inguinal hernia repairs are performed worldwide. The incidence of finding an appendix in the hernia sac is, on average, below 1%, with values varying in specialized literature between 0.19 and 1.7% [3,4,5].

Concomitant Amyand inguinal hernia with acute appendicitis is even rarer, 0.1% of all appendicitis cases. The pathology is more common in children, more frequently affecting the male sex, and in most cases, it is located on the right side. Placement on the left side corresponds to excessive mobility, intestinal malrotation, a floating cecum, or situs inversus. 

The widely accepted definition of Amyand’s hernia is the presence in the inguinal hernia sac of the appendix, with or without inflammatory changes.

Although some authors claim that acute intra-saccular appendicitis is necessary to define Amyand’s hernia, most also accept the situation of a non-inflamed appendix. Moreover, the therapeutic protocols also include the variant with a healthy appendix.

The pathophysiology of this condition is debatable. Two hypotheses are widely accepted. The first states for the vermiform appendix evolving towards the hernia sac through the persistence of the peritoneo-vaginal canal. This hypothesis explains the much higher frequency of Amyand’s hernia in children [6].

In adults, bands of fibrous tissue have been found connecting the hernia sac to the testis, which may act as a guiding mechanism for appendicular protrusion. In 0.13% of all cases of Amyand’s hernia, the appendix shows inflammatory changes. Progression to appendiceal perforation can evolve into serious complications for the patient with increased mortality [7,8,9,10,11,12,13].

In pediatric patients, especially those of young age, obtaining a historical and clinical diagnosis can be challenging. The approach regarding the preservation of the appendix in situations where it does not reveal inflammatory changes should be directed towards the conservation of this organ. Considering that, usually in the pediatric age group, the etiology of hernia is attributed to the persistence of the peritoneo-vaginal canal, the use of a mesh for hernia repair is not a first-line option [14,15].

The method of producing appendicitis in the inguinal hernia sac is not fully elucidated. The evolution of the hernia towards incarceration, which will cause inflammation of the appendix, is one of the proposed mechanisms. Another hypothesis is that the contraction of the abdominal muscles causes appendicular obstruction or that intrasaccular adhesions predispose to subsequent incarceration.

A vicious circle appears, consisting of appendicular inflammation with evolution towards hernial irreducibility, which will determine the accentuation of the inflammatory changes of the appendix [16,17,18,19,20,21,22,23].

However, most frequently, this condition is asymptomatic, the appendix being discovered without inflammatory changes during inguinal hernia repair (Figure 1).

When symptoms occur, they are specific to a strangulated hernia rather than acute appendicitis, with groin pain, acute irreducibility, and sometimes nausea and vomiting. A specific element, however, is the character of the pain—it is not continuous or suddenly installed but rather insidious and colicative. The local physical examination shows swelling of the groin, tenderness, cellulite, or even local necrosis in neglected forms. The diameter of the hernia defect has its own role when it comes to peritonitis limitation.

Thus, a small diameter will limit the peritonitis at the saccular level, and a large diameter will allow the process to advance to generalized peritonitis [7,8,11].

The differential diagnosis should be performed with a strangulated hernia, orchiepididymitis, acute hydrocele, testicular tumor or testicular torsion, Richter’s hernia, and acute appendicitis.

In complicated forms, laboratory analyses may show increased inflammatory markers or even biohumoral changes in the context of sepsis.

According to the previous mentions, the diagnosis is usually intraoperatively made. This is also due to the fact that in the situation of a condition that mimics a strangulated hernia, the vast majority of surgeons do not request additional investigations, preferring surgical intervention. Situations where the diagnosis of Amyand’s hernia was established only on the basis and in accordance with the clinical examination are rare [18,19,22,23].

Ultrasound and tomography are imaging diagnostic methods. During the ultrasound examination, the existence of an intrasaccular structure can be highlighted, and during the CT, the presence of a tubular structure that originates at the base of the cecum and which may or may not show CT changes specific to inflammation [8,11,24,25].

Although it is not customary to perform a preoperative imaging investigation such as ultrasound or CT scan in chronic cases because of additional costs that are involved, these investigations prove to be particularly useful, especially in acute situations. A preoperative diagnosis of Amyand hernia can help while making the differential diagnosis between strangulated hernia and simple incarceration. It can also provide details regarding the content of the sac or about the strangulated organ (appendiceal changes, intestinal vascular ischemia, or intrasaccular intestinal perforations). These data provide the surgeon with a spectrum for more efficient preoperative preparation and can guide him while choosing the type of surgical approach. Adequate preoperative preparation and short-duration surgical intervention will impact the patient’s outcome [26].

The therapeutic, surgical option is often determined by the type of presentation (urgent or scheduled), the nature of the symptoms, and the preoperative diagnostic suspicion. Acute forms with symptoms specific to strangulation often benefit from open surgery. Forms with nonspecific manifestations, with debatable signs of hernia strangulation, can be approached laparoscopically [27].

Over time, several classifications have been proposed for Amyand’s hernia. The most used are that of Losanoff and Basson (Table 1), as well as Rikki’s classification, which represents a modified form of the first one.

In Rikki’s classification, the fifth type of Amyand hernia is added, the intrasaccular protrusion of the appendix being realized not through an inguinal hernia but through a post-incisional one [10]. At the same time, it was also proposed to make a classification of the vermiform appendix in the hernia sac. Thus, we have type A, without inflammatory changes, type B, with inflammatory changes, and a perforated appendix in type C [28,29].

Late diagnosis and treatment of the condition can lead to a mortality of between 14 and 30%, mainly due to the associated sepsis. Mortality drops to 5.5% if treatment is started early [3,6,7,8,30].

## 2. Materials and Methods

### 2.1. Literature Review for Open Approach

The rare frequency of this association did not allow standardized surgical management regarding the parietal repair and the attitude towards the intrasaccular appendix. This topic is a controversial one, and in order to see which surgical tactic is preferred regarding the use of a textile allograft in open surgery following an appendicectomy, a thorough literature search and analysis were carried out to identify optimal management strategies.

Since the most frequent reports regarding this type of pathology are case reports, we studied single-patient case reports or case series reports. The review of the database also highlighted reviews that were separately analyzed. These ones mostly present general data regarding the condition and less data concerning surgical management.

As such, this study aims to provide additional data regarding the therapeutic strategy of the controversial sections, which surgical tactic is preferred regarding the use of a textile allograft in open surgery following an appendicectomy, and what the attitude is towards a healthy appendix. This article adhered to the PRISMA guidelines for reporting systematic reviews, and the PRISMA checklist was completed for the manuscript and abstract.

We conducted a review of the literature using the PubMed database (ncbi.nlm.nih.gov accessed on 1 July 2022), utilizing a search syntax composed of the following terms: “Amyand hernia” and “mesh,” obtaining 70 results (Figure 2). The inclusion criteria for the selected studies were articles written in English on adult cases (over 18 years of age) with the presence of vermiform appendix in the hernia sac and the use of a prosthetic open surgery procedure as a solution to repair the parietal defect. The exclusion criteria were represented by articles written in a language other than English, pediatric patients, articles where neither full text nor abstract was found, non-mesh procedures, other types of hernia, and letters to editors. Literature reviews and the laparoscopic approach were studied separately. This review was performed in accordance with the PRISMA (Preferred Reporting Items for Systematic Reviews and Meta-Analyses) guidelines.

Data interpretation can be subject to bias, so some clarifications are necessary regarding certain aspects of the study. Concerning the type of hernia at the time of hospitalization for those that were not specified, we considered them to be programmed cases in which no clinical signs of incarceration or strangulation were identified. In the situation where there were presentations with a painful but reducible inguinal hernia and the appendix with inflammatory changes, we considered that the appendicitis was of the primary intrasaccular type and not secondary to a strangulation mechanism.

In the reports in which the Losanoff and Basson classification was not precisely specified, the classification was made according to the data presented in the text on the macroscopic or microscopic examination of the appendix. Taking into account the frequency of using standard meshes when practicing the repairing process of the abdominal wall, we considered this type of prosthesis as being used in cases where the type of mesh was not specified.

At the end of the search process and after excluding non-compliant data, 32 articles totaling 55 patients were obtained. The obtained data are summarized in Table 2.

### 2.2. Literature Review for Laparoscopic Approach

Considering the increased tendency of the laparoscopic approach of inguinal hernias and also of the condition of acute abdomen, respectively the controversial aspects of the surgical treatment for this type of approach, we carried out a review of the specialized literature in an attempt to extract the predilection towards a certain sequence of surgical steps (Figure 3). We analyzed the relationship between the type of appendix, its surgical approach, and the choice of hernia treatment by using the PubMed database and entering the words “Amyand hernia” and “laparoscopy” into the search engine. Pediatric cases were not taken into account. Other hernia types, articles in languages other than English, and articles with insufficient data were excluded.

After analyzing the data obtained during the research and eliminating ineligible articles or cases, 16 patients and 14 articles were identified, the results being summarized in Table 3:

## 3. Results

### 3.1. Results of the Open Approach Study

The incidence of Amyand hernia is significantly higher for males, 92.73% (*n* = 51), compared to females, 7.27%, the average age for males being 56.26 years and for females 66.75 years. Almost 1/3 (34.55%) of the studied specimens presented acute manifestations and were hospitalized as urgent cases, 2/3 (65.45%) being scheduled cases for hernia repair with the accidental discovery of the appendix at the saccular level.

#### Surgical Findings, Management, and Outcomes in Open Surgery

Amyand hernia with non-inflamed vermiform appendix (type A) was intraoperatively detected in 76.36% of patients, type B in 20%, and type C with associated peritonitis in 3.64%. The presence of six cases of admission through the emergency service in patients with a type A appendix is explained by the existence of pain symptoms that overlapped with a strangulated hernia and not by the existence of an acute intrasaccular appendicitis. Establishing the pathophysiological sequence of strangulation with appendicular inflammatory changes vs. acute appendicitis with intrasaccular inflammatory changes is almost impossible. An indicative element would be the existence of a clinical sequence, such as irreducibility followed by local pain pleading for strangulation as a primary mechanism or finding imprinted marks on the digestive segment located in the hernia sac during intraoperative examination. The studied cases did not present descriptive details of this type.

The reduction of the appendix in the peritoneal cavity was practiced in 47.27% of the studied cases, with appendectomy being practiced in 52.73% of the cases (*p* = 0.0018). Regarding type B or C appendixes, the attitude towards the appendix was clearly named as being appendectomy, while concerning appendix type A, no consensus was observed. Thus, in 61.9% (n = 26) of patients, surgeons opted for the reduction of the appendix in the peritoneal cavity, while for 31.1% of cases (n = 16), an appendectomy was the procedure of choice.

The repair of the parietal defect using allografts without special characteristics was the main option, being applied in 92.73% of patients, and the remaining 7.27% benefitted from allografts with special characteristics (absorbable, large porous, or biodegradable).

Evaluating the relationship between the type of prosthesis (standard or with special characteristics) and the type of appendix, it was observed that standard allografts were mainly used for every type of appendix, i.e., in 97.62% of those with type A, 72.73% of those with B-type, and in all of those with C-type appendix (OR = 0.08, 95% CI = 0.01–0.87, z-score = 2.08, *p* = 0.038). The use of allografts with special characteristics was significantly more frequent in the group of patients with an inflamed appendix (27.27%) compared to those with a normal appendix (2.38%).

Two postoperative complications (2.64%) were recorded, both after appendectomy and hernia repair by using standard mesh: a wound infection for a type A appendix for emergency surgery and one postoperative seroma for a type B appendix also for an emergency presentation, but without identifying the statistical significance related to this sequence of events (*p* > 0.05).

### 3.2. Results for the Laparoscopic Study

Amyand hernia type was detected more frequently in males, 81.25%, with females being affected in a proportion of 18.75%. The average identified age was 55.56 years, 56.15 for men and 53 for women.

Type A Amyand’s hernia presents without inflammatory changes in the groin. In type B Amyand’s hernia, septic changes are limited to the hernia sac, while type C Amyand’s hernia entails sepsis spreading beyond the hernia sac that can develop into acute appendicitis and other abdominal lesions [66].

The detection of an appendix in the hernia sac was predominantly identified in patients admitted as urgent cases (73.33%), more frequently for type B and rarely for appendicular type A (Table 4). Statistical significance was recorded between the type of presentation and the type of appendix (*p* = 0.003).

Analyzing by age groups, a higher frequency of the pathology was observed in the 45–70 age group (77.77%), with the predominance of type B in this age category (Figure 4).

#### Surgical Findings, Management, and Outcomes of Laparoscopic Interventions

The laparoscopic appendectomy was performed in 68.75% (n = 11) of all cases. In 25% (n = 4) and 6.25%, respectively, conversion and open appendectomy were performed.

Type A variant of the appendix-associated laparoscopic appendectomy in 12.5% (n = 5) of cases, preservation of the appendix being done for 18.75% (n = 3) of the total number of patients. In the type A appendix group (n = 5), the laparoscopic appendectomy was performed in 40% of cases (OR = 0.15, 95%CI = 0.01–1.56, z statistic = 1.59, *p* = 0.094), conservation being done in 60% of cases (OR = 0.07, 95% CI 0–1.02, z statistic = 1.95, *p* value = 0.029).

In patients with the type B variant of the appendix, representing half of the studied cases, a laparoscopic appendectomy was practiced in 87.5% of cases (n = 7), representing 43.75% of the total interventions addressed to the appendix. In only one case, conversion to an open procedure was performed. For cases with a type C appendix, laparoscopic appendectomy was preferred in n = 2 cases, with conversion being practiced in only one case.

Regarding the parietal defect, it is noted that in one case, due to generalized peritonitis, the treatment of the hernia defect was abandoned. In 18.18% of cases, the hernia repair was performed through tissue procedures in open surgery; in 27.27% of cases, textile allografts were used in the open variant, while the laparoscopic tissue approach was performed in 27.27% of cases. The laparoscopic with surgical mesh approach was used in 36.36% of the studied cases. For 42.8% of cases with the mesh hernia repair, it was preferred for the hernia treatment to be carried out in a second intervention.

The non-inflamed appendix is more frequently associated with TAPP as a treatment procedure for the parietal defect, with 50% of patients with type A appendix being operated on using the same procedure (OR = 0.07, 95%CI 0–1.02, z statistic = 1.95, *p* value = 0.029).

Non-mesh hernia repair was more frequently preferred in type B appendixes (66.66%), with TEP being used in two cases with type B and one with type A appendix.

Two-stage surgical interventions were used for type C appendixes (66.66% delayed Liechtenstein and 33.33% delayed non-mesh hernia repair). One case of type B appendix underwent an appendectomy and TAPP intervention in a secondary time.

The laparoscopic appendectomy associated with TAPP in the same operative time was highlighted in 9.09% of cases (OR = 0.07, 95% CI 0–1.02, z statistic = 1.95, *p* = 0.029). Cases that benefited from appendix preservation associated with TAPP were in the proportion of 75% (*p* = 0.008). The outcome was favorable in all patients, with no major complications being reported.

## 4. Discussions

The therapeutic strategy for Amyand’s hernia remains controversial because this type of condition brings additional elements to a standard inguinal hernia. The existence of a non-essential organ in the hernial sac, which shows septic evolution either at the time of discovery or over time, the immunological, regulatory role of the intestinal microbiota, the involvement in autoimmune diseases, and the potential of evolution towards neoplasia make the therapeutic option a complex one.

The surgeon must take into account, in addition to the previously mentioned, the fact that he must also obtain efficient, tension-free parietal restoration without local complications and with the lowest recurrence rate.

If, in the case of an inflamed or perforated appendix, appendectomy is unanimously accepted for type A, the surgical option is still under debate.

The vast majority of surgeons propose preservation of the appendix, reduction of the hernia, and treatment of the parietal defect with textile allografts. This option is mainly related to the desire to avoid a septic time in an operation where a prosthetic solution is usually preferred. Appendectomy in the form of Amyand’s hernia with a healthy appendix should be reconsidered in the perspective of new reports regarding the function and importance of the appendix in regulating the colic bacterial flora, protection against inflammatory bowel diseases, and Parkinson’s disease. At the same time, the appendix can be used in exceptional cases, if needed, as a substitute material for extrahepatic bile ducts, urinary diversions, or appendicostomy [67,68,69,70].

However, there are also authors who indicate an appendectomy in any Amyand hernia. The reasoning is related to the risk of acute appendicitis over time, which is why, through the appendectomy, they propose the elimination of this type of differential diagnosis in an upcoming right iliac fossa pain syndrome.

However, we notice that most of those who propose this type of attitude are those who report pediatric patients with Amyand’s hernia, a situation in which an appendectomy has a stronger indication. The reason is the higher risk of developing acute appendicitis in the first decades of life. There are also authors who postulate that intraoperative manipulation of the appendix could cause inflammation.

These cases are rare, and we consider that an adequate manipulation, without injuries or ruptures of the appendix, does not cause such a situation [8].

Appendectomy, even in the case of a healthy appendix, is mandatory in an Amyand hernia located on the left side due to the risk of a future acute appendicitis and wrong diagnosis with the possibility of delayed or erroneous treatment [8,57,71,72].

As a general rule, after an appendectomy, an inflamed or perforated appendix (type B and C) requires the repair of the defect to be made in tissue variant, without the use of allograft. The reasons relate to the increased risk of wound infection, sepsis, allograft rejection, and appendiceal stump fistula or hernia recurrence [7,33,34,35,73].

However, some authors also describe the use of “mesh” in acute appendicitis without complications. They claim that sustained antibiotic therapy and postoperative drainage relieve the patient of postoperative complications. Moreover, in our study, there is only one complication associated with appendectomy for an inflamed appendix [73,74].

In a large study that included 72 adult patients with type A, Manatakis et al. noted the option to preserve the appendix in 35% of cases, with allograft mounting in 88% of cases from this group. Appendectomy for the group with non-inflammatory appendices was preferred in 65% of cases, associating mesh repair in the quantum of 62%. In acute cases of 142 patients who underwent appendectomy, the repair of the parietal defect was performed with allograft in 19% of cases, with tissue procedures being used in 81% of patients. The authors did not report local complications in situations where the mesh was mounted regardless of the type of appendix [75].

Papaconstantinou et al. reported a percentage of 82.7% of mesh repair for elective surgical interventions with a non-inflamed appendix and 26.2% for cases with an appendix with inflammatory changes. As in the previous study, no postoperative complications determined by mesh utilization were recorded regardless of the type of the appendix or the elective or urgent nature of the cases [76].

The recommendations of the World Society of Emergency Surgery (WSES) for the “clean-contaminated surgical field” (CDC wound class II) do not contraindicate the mounting of textile allografts, both standard and with special properties. This attitude in association with antibiotic therapy does not have a negative impact on the evolution of septic complications or relapse [77,78].

In our review, we note the increased frequency of appendectomy for situations with a non-inflamed appendix, as well as the preferred use of meshes without special biological characteristics. There is also the existence of a personal character in terms of the attitude towards the appendix, the surgeon being the one who gets to decide on a certain tactic. There are authors who report preservation of the appendix in all hernias with appendix type A and reports in which appendectomy was unquestionable.

It is difficult to draw any firm conclusions regarding the attitude towards the appendix and mesh placement, considering that many of the authors (representing 42.27% of the patients) preferred the preservation of the non-inflamed appendix. For type B and C appendices, where appendectomy is the rule, the controversies concern the allograft or tissue procedure option. Thus, some authors performed procedures with textile allograft placement even for the variant with an inflamed appendix. However, if we compare this with the number of reports with laparoscopic studies in which tissue procedures were used (n = 14), we can conclude that a local inflammatory/septic process, especially stage C, basically contraindicates the placement of a mesh. Another increasingly common approach is the laparoscopic and classical mixed approach. In this way, laparoscopic appendectomy is performed, and later, through open surgery, the hernia is solved with a prosthetic repair. However, this attitude depends on two aspects: having the diagnosis of Amyand hernia with a healthy or inflamed appendix preoperatively established (rare case), or a preoperative diagnosis of acute abdomen or acute appendicitis, in which a laparoscopic approach is chosen and the Amyand hernia is thus detected. In these situations, after the laparoscopic stage, hernia repair can be performed in open surgery with surgical mesh.

There are situations in which surgeons prefer to perform a laparoscopic appendectomy per primam and then, a few months later, use laparoscopic or open hernia treatment. Laparoscopic fitting of textile allograft in an Amyand hernia with acute appendicitis, even after resection of the sac and the use of low weight and large porous mesh, is considered inappropriate most of the time.

Another emphasized situation was the preference for the laparoscopic approach while practicing appendectomy in cases with an Amyand hernia, despite the absence of inflammatory changes.

The treatment of the abdominal wall defect was not uniform, with the laparoscopic without mesh option being preferred, especially for types I and II of appendices. Laparoscopic placement of a mesh was practiced more frequently for type I; for the other situations, a second surgical intervention at a distance from the first one was preferably chosen.

We can observe the increasing frequency regarding the placement of textile allografts, even in the situation of a septic time or in the presence of a local inflammation. The preservation of the intrasaccular non-inflamed appendix is practiced more frequently, with the ratio between this attitude and appendectomy being almost equal [11,63,79].

The rare situations in which Amyand’s hernia is associated with appendicular neoplasia require a completely different approach centered on the treatment of the cancer. Finding the proper solution for repairing the hernia defect is taking a setback, and the intervention is being oriented to obtain an optimal onco-surgical result. The association of appendicular neoplasia with an Amyand hernia is very rare (9 reported cases), the therapeutic modality for these cases being represented by an appendectomy, which underlines the fact that the existence of an appendicular carcinoma was revealed by the histopathological examination and not by the clinical suspicion accompanying the intraoperative macroscopic examination [79].

Depending on the histological type, the local or systemic extension, and the response to the oncological therapy, the surgical approach can be reduced to an appendectomy, or more extensive resections can be performed in a secondary time.

More recently, the treatment of Amyand’s hernia can also be performed by robotic surgery. However, this type of approach is an elective one and requires, in addition to specialized instrumentation, high costs and a definite preoperative imaging diagnosis [80].

Multiple aspects of the therapeutic management of inguinal hernia are still under debate, the main points of controversy being the use of the meshes in forms with acute appendicitis and the attitude towards the non-inflamed appendix. For the first intriguing aspect, we consider it a matter of decision, depending on the surgeon’s experience with such situations, the local conditions, and the degree of appendicular inflammation.

## 5. Conclusions

Amyand’s hernia is rarely encountered in surgical practice, its preoperative diagnosis being a challenge. The treatment of this condition can be carried out in different ways. The association of the appendectomy with the placement of a textile allograft is feasible even in the case of an inflamed appendix, with complications related to this option being rarely met. However, in the interim, we consider it useful to follow standardized surgical principles at the expense of innovative or new approaches that have not yet proved their effectiveness in large studies.

## Figures and Tables

**Figure 1 diagnostics-13-03534-f001:**
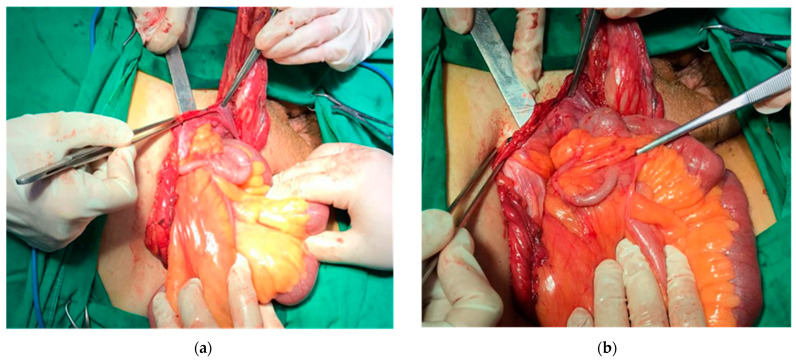
(**a**,**b**)—Personal collection—Type A appendix in an Amyand hernia in open approach.

**Figure 2 diagnostics-13-03534-f002:**
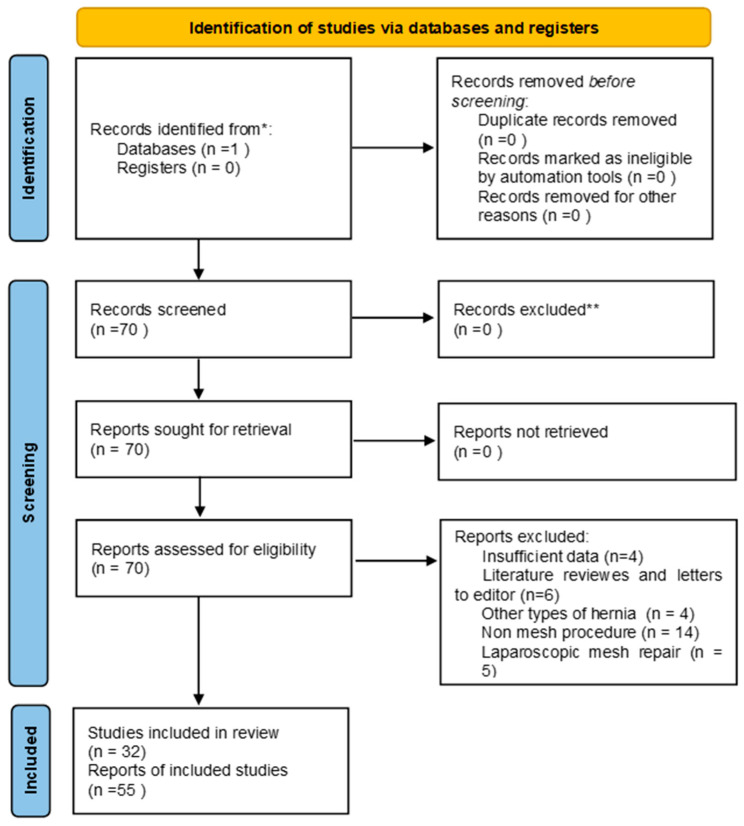
PRISMA diagram for the open approach study.

**Figure 3 diagnostics-13-03534-f003:**
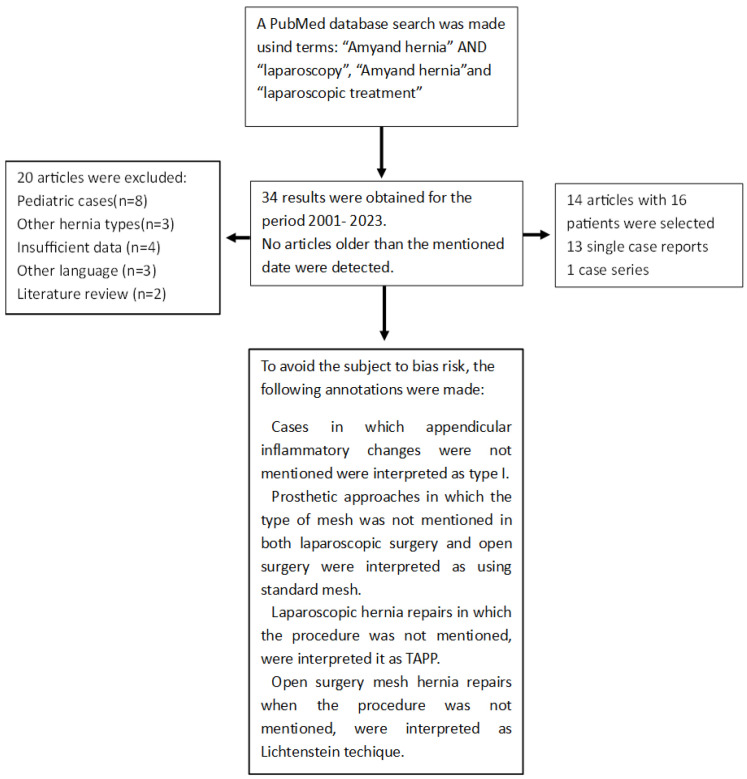
Chart for laparoscopic study.

**Figure 4 diagnostics-13-03534-f004:**
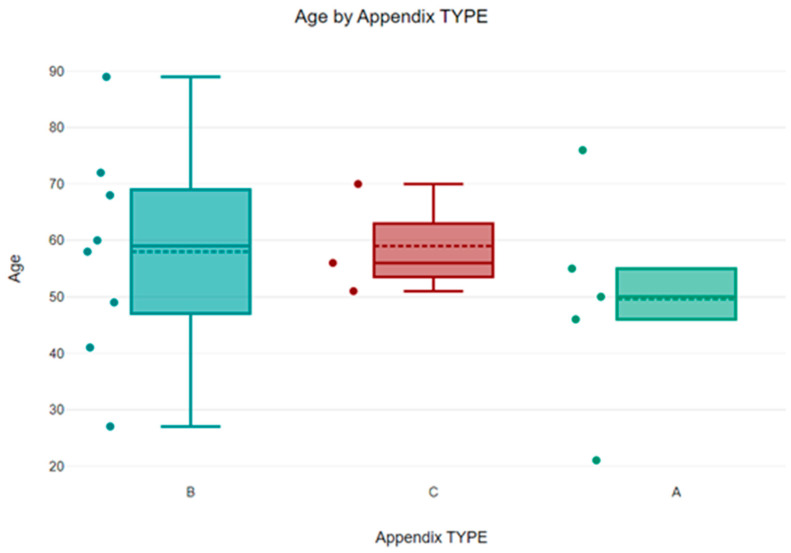
Appendix type distribution by age groups. Blue for type B, red for type C and green for type A.

**Table 1 diagnostics-13-03534-t001:** Losanoff and Basson classification.

Losanoff and Basson Classification	Description	Surgical Management
Type I	Normal appendix within an inguinal hernia	Hernia reduction, mesh repair, and appendectomy only in young patients
Type II	Acute appendicitis within an inguinal hernia with no abdominal sepsis	Appendectomy, primary non-mesh hernia repair
Type III	Acute appendicitis within an inguinal hernia with abdominal wall or peritoneal sepsis	Laparotomy, appendectomy, non-mesh hernia repair
Type IV	Acute appendicitis associated with related or unrelated abdominal pathology	Appendectomy through hernia incision or laparotomy with management of concomitant disease

**Table 2 diagnostics-13-03534-t002:** Review of literature regarding open approach in patients with Amyand hernia.

Authors	Year	No. Cases	Appendix Type	Appendix Management	Type of Mesh	Postoperative Outcome
Logan MT et al. [31]	2001	1	A	Reduction in the abdominal cavity	Standard	Uneventful
Priego P et al. [32]	2005	3	A	Appendectomy	Standard	Wound infection
Sharma H et al. [33]	2007	11	A	Reduction in the abdominal cavity	Standard	Uneventful
Torino G et al. [34]	2007	1	C	Appendectomy	Standard	Uneventful
Inan I et al. [35]	2009	2	B	Appendectomy	Standard	Uneventful
Karatas A et al. [36]	2009	1	A	Reduction in the abdominal cavity	Standard	Uneventful
Psarras K et al. [25]	2011	2	A	Reduction in the abdominal cavity	Standard	Uneventful
Burgess PL et al. [37]	2011	1 1	B A	Appendectomy Appendectomy	Absorbable mesh Absorbable mesh	Uneventful Uneventful
Ranganathan G et al. [38]	2011	1	B	Appendectomy	Standard	Uneventful
Sengul I et al. [39]	2011	1	A	Appendectomy	Standard	Uneventful
Ali SM et al. [40]	2012	1 1	B A	Appendectomy Appendectomy	Large pore-sized mesh Large pore-sized mesh	Uneventful Uneventful
Quartey B et al. [41]	2012	1	A	Appendectomy	Standard	Uneventful
Yıldız M et al. [42]	2012	1	B	Appendectomy	Biological mesh	Uneventful
Junaid J et al. [43]	2012	1 1	A A	Appendectomy Appendectomy	Standard Standard	Uneventful Uneventful
Al Maksoud AM et al. [44]	2015	1	A	Reduction in the abdominal cavity	Standard	Uneventful
Singhal S et al. [24]	2015	1	A	Reduction in the abdominal cavity	Standard	Uneventful
Morales-Cárdenas A et al. [45]	2015	1	A	Appendectomy	Standard	Uneventful
Michalinos A et al. [46]	2015	3	A	Appendectomy	Standard	Uneventful
Reilly DJ et al. [47]	2015	1	B	Appendectomy	Standard	Uneventful
Goyal S et al. [48]	2015	1	A	Reduction in the abdominal cavity	Standard	Uneventful
Kose et al. [49]	2017	5	A	Appendectomy	Standard	Uneventful
Shaban Y et al. [4]	2018	1	A	Appendectomy	Standard	Uneventful
Kosmidis C et al. [50]	2018	1	A	Appendectomy	Standard	Uneventful
Okita A et al. [51]	2020	1	A	Appendectomy	Lightweight mesh	Uneventful
Kakodkar P [52]	2020	1	A	Appendectomy	Standard	Uneventful
Tsalis K et al. [53]	2021	1	A	Reduction in the abdominal cavity	Standard	Uneventful
Khalid H et al. [54]	2021	1	B	Appendectomy	Standard	Uneventful
Elgazar A et al. [55]	2021	1	A	Reduction in the abdominal cavity	Standard	Uneventful
Regmi BU et al. [56]	2022	1	A	Appendectomy	Standard	Uneventful
Heo TG et al. [57]	2022	1	A	Reduction in the abdominal cavity	Standard	Uneventful
Bawa A et al. [58]	2023	1	A	Reduction in the abdominal cavity	Standard	Uneventful
Corvatta FA et al. [59]	2023	1	A	Appendectomy	Standard	Uneventful

**Table 3 diagnostics-13-03534-t003:** Review of literature regarding laparoscopic approach in patients with Amyand hernia.

Authors	Year	Appendix Type	Surgical Approach for Appendix	Hernia Repair—No Mesh	Hernia Repair—Mesh
Assad MA et al. [60]	2023	Type B	Laparoscopic appendectomy		TEP
Gupta AK et al. [61]	2020	Type C (generalized peritonitis)	Laparoscopic converted to midline laparotomy and appendectomy		Delayed Lichtenstein
Garagliano JM et al. [62]	2020	Type C	Laparoscopic appendectomy	Open approach no mesh hernia repair	
Han SH et al. [63]	2019	Type B	Laparoscopic appendectomy		TEP 3 months later
Syllaios A et al. [64]	2019	Type B	Laparoscopic appendectomy		Laparoscopic mesh repair
Muroya D et al. [65]	2019	Type A	Laparoscopic appendectomy		TEP
Akaishi R et al. [11]	2018	Type C	Laparoscopic appendectomy		Open surgery hernia repair 1 month later
Abdulla S et al. [12]	2017	Type B	Laparoscopic appendectomy	Open approach no mesh hernia repair	
Sahu D et al. [13]	2015	Type A Type A Type A	Laparoscopic appendix reduction Laparoscopic appendix reduction Laparoscopic appendix reduction		TAPP TAPP TAPP
Bailon-Cuadrado M et al. [20]	2016	Type B	Exploratory laparoscopy, open appendectomy		Open surgery hernia repair
Yagnik VD et al. [21]	2011	Type B	Laparoscopic appendectomy		Open surgery hernia repair 1 month later
Elias B et al. [22]	2011	Type B	Laparoscopic appendectomy	Laparoscopic no mesh hernia repair	
Mullinax JE et al. [23]	2011	Type B	Laparoscopic appendectomy	Laparoscopic no mesh hernia repair	
Bamberger PK et al. [29]	2001	Type A	Laparoscopic appendectomy	Laparoscopic no mesh hernia repair	

**Table 4 diagnostics-13-03534-t004:** Frequency of appendix type according to the type of presentation.

		Appendix Type			
		B	C	A	Total
Presentation	Emergency	46.67%	20%	6.66%	73.33%
	Elective	0%	0%	26.67%	26.67%
	Total	46.67%	20%	33.33%	100%

## Data Availability

Data are contained within the article.

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
