# Peer review of "Controversies Regarding Mesh Utilisation and the Attitude towards the Appendix in Amyand’s Hernia—A Systematic Review"

_diagnostics, 2023, doi:10.3390/diagnostics13233534_

Round 1

Reviewer 1 Report

Comments and Suggestions for Authors

The paper discusses the diagnostic challenges, therapeutic approaches, and surgical ethics associated with this condition, aiming to offer insights and trends for the management of such hernias, even in the absence of large-scale studies or extensive reviews.

Here are a few comments that should be addressed before acceptance. 

How can advances in medical imaging technology contribute to more accurate and timely diagnosis of this condition, reducing the likelihood of intraoperative surprises?

What factors influence the choice of surgical approach in managing this rare condition, and how do they impact patient outcomes?

What are the most reliable diagnostic methods for identifying inguinal hernias containing the vermiform appendix, especially in cases without inflammatory changes?

What are the differences in the clinical presentation, diagnosis, and management of this condition in pediatric patients compared to adults, and how can these differences inform treatment decisions?

 The language usage throughout this paper needs to be improved; the author should do some proofreading on it.

 The introduction section can be extended to add the issues in the context of the existing work

More clarifications and highlights about the research are in the related works section. I suggest to discuss the following studies:

-A Novel Light U-Net Model for Left Ventricle Segmentation Using MRI

-A Comparative Analysis of Optimization Algorithms for Gastrointestinal Abnormalities Recognition and Classification Based on Ensemble XcepNet23 and ResNet18 Features

-A Comprehensive Survey on Quantum Machine Learning and Possible Applications.

Author Response

Esteemed Reviewer,

We are providing an additional response, point by point, regarding the changes you suggested for our article:
- Points 1, 2, 3 and 4 have been addressed in the text, in individual paragraphs, with corresponding references. The responses have been presented in detail in the previous message.

-The manuscript has been revised, and the identified errors, including punctuation errors, have been corrected.

-The introduction section has been expanded.

-The studies you mentioned are indeed interesting and some of them have been incorporated into the text to add value to the material.

- Additionally, the references and text structure have been updated to meet MDPI standards.

Once again, we would like to express our gratitude for your effort in evaluating our material.

Reviewer 2 Report

Comments and Suggestions for Authors

well done and interesting paper to read and to know about

Author Response

Thank you for the effort you have put into the review process of our article!

Reviewer 3 Report

Comments and Suggestions for Authors

A nice and thorough metanalysis on the subject. There are some issues related to the references that should appear in order in text;for instance line 68 after 10 should be 11 instead of 54. The type of appendicitis (A, B,C) should be explained in the text like you did for the type of hernia.

Comments on the Quality of English Language

Minor revision required

Author Response

Thank you very much for taking the time to review our manuscript!

The article's references will be adjusted during the minor revision process conform to your suggestions and further statements regarding the classification of the appendix will be incorporated within the paragraphs of the paper.